# Ten Actions to Counteract Vaccine Hesitancy Suggested by the Italian Society of Hygiene, Preventive Medicine, and Public Health

**DOI:** 10.3390/vaccines10071030

**Published:** 2022-06-27

**Authors:** Claudio Costantino, Caterina Rizzo, Roberto Rosselli, Tatiana Battista, Arianna Conforto, Livia Cimino, Andrea Poscia, Daniel Fiacchini

**Affiliations:** 1Department of Health Promotion Sciences, Maternal and Infant Care, Internal Medicine and Medical Specialties (PROMISE) “G. D’Alessandro”, University of Palermo, 90127 Palermo, Italy; arianna.conforto@unipa.it (A.C.); livia.cimino@unipa.it (L.C.); 2Department of Traslational Research and New Technologies in Medicine and Surgery, University of Pisa, 56122 Pisa, Italy; caterina.rizzo@unipi.it; 3Local Health Unit of Genoa (LHA 3) 3, 16129 Genoa, Italy; roberto.rosselli@asl3.liguria.it; 4Prevention Department, Local Health Unit of Taranto, 74121 Taranto, Italy; tatiana.battista@asl.taranto.it; 5Public Health Department, Asur Marche AV2, 60100 Ancona, Italy; andrea.poscia@sanita.marche.it (A.P.); daniel.fiacchini@sanita.marche.it (D.F.)

**Keywords:** vaccine hesitancy, counteracting hesitancy, standardized practices, community of practices, collaborating research, scientific societies, health planning tools, monitoring systems

## Abstract

Vaccine hesitancy (VH) is one of the main causes of the widespread decline in vaccination coverage and has become the subject of ongoing debate among public health professionals. The present commentary is a “decalogue” of strategic actions to counteract vaccine hesitancy for public health professionals that comes from the cognitive and formative path put in place by the “Communication in Public Health” working group (WG) of the Italian Society of Hygiene, Preventive Medicine, and Public Health. From the establishment of a national, multidisciplinary WG on VH to the activation of a national monitoring/surveillance system on vaccine hesitancy, several proposals are discussed. The identification and dissemination of good practices and tools to counteract and understand vaccine hesitancy, interdisciplinary training on vaccine hesitancy and on risk communication, community engagement and infodemiology, the inclusion of effective interventions to counteract vaccine hesitancy within the National Immunization Plan (NIP), the promotion and growth of a community of practice and research in the field of vaccine hesitancy, collaborations between scientific societies, and knowledge from the behavioral sciences represent other actions recommended in the present commentary. The present document suggests ten undeferrable strategies that could be implemented at the national and local levels in Italy, and that could be borrowed by other European countries in order to counteract vaccines hesitancy with a systematic and organic approach.

## 1. Introduction

The phenomenon of so-called “vaccine hesitancy” (VH) is considered to be one of the main causes of the widespread decline in vaccination coverage and has become the subject of an ongoing debate among public health professionals. Specifically, the term indicates a delay in the acceptance or refusal of vaccinations by parents or individual citizens despite the availability of vaccination services that offer free vaccinations.

It is a global, complex and ever-changing phenomenon, and it represents one of the most important criticisms in public health today [1].

A first attempt at framing this multifaceted situation was proposed in 2000 by a study taken up by the European Center for Disease Control and Prevention (ECDC), which distinguishes patients into different categories:Hesitant: concerned about the safety of vaccines and unsure about needs, procedures and timetableDisinterested: with little awareness of vaccination (considered a low priority) and an inadequate perception of the risk of preventable diseasesExcluded: disadvantaged with limited or difficult access to treatment for social, economic and integration reasonsAnti-vaccinationists: with an attitude of rejection and active resistance due to personal, cultural and religious convictions.

One of the main factors contributing to the development of vaccination hesitancy is concern about the safety of vaccines, encouraged by new media and social media, which have profoundly changed the way the general population accesses health information, as well as the heterogeneity and truthfulness of its contents [2,3]. In this way, anti-vaccination groups have found using the Internet to be an effective way of defending their ideas [2,3,4].

The mass media’s emphasis on the hypothetical side effects of vaccines has triggered waves of misinformation on the safety of vaccines, the so-called infodemic, mainly concerning long term side effects and ADE, the toxicity of adjuvants and preservatives, and the weakening of the immune system [5,6].

The WHO’s Strategic Advisory Group of Experts (SAGE) on immunization notes that misinformation about vaccine safety has had a negative effect on vaccination campaigns [7]. According to the WHO, the acceptance of vaccination is the result of a complex decision-making process that can potentially be influenced by a wide range of factors. In 2012, SAGE analyzed a number of conceptual models for identifying these factors, which are referred to as determinants [7]. There is a wide variety of determinants of vaccine hesitancy. Many historical, social, cultural, environmental, economic, political, and institutional factors can influence vaccination choices [7]. Then there are personal perceptions, beliefs about vaccines, influences from the social environment or religious choices, and socio-demographic characteristics [7].

During the COVID-19 pandemic and after the start of the greatest immunization campaign of the history, challenging VH and new determinants generated among different populations (health care professionals, elderly, fragile people, pregnant women, etc.) became a public health priority [8,9,10]

One of the best strategies for dealing with VH was found to be one based on structuring interventions around the empowerment of general population. Italy has recently been a great example of adherence to its COVID-19 vaccination campaign, with 90.2% of the population older than 12 y.o. receiving a complete vaccination cycle [11]. However, Italy has not always been a good example in terms of pediatric vaccinations during past few years.

As an example, in 2017/2018, Italy faced a measles epidemic due to low vaccination coverage in previous years, which resulted in more than eight thousand cases (74% > 15 years of age) and 15 deaths [12]. Satisfactory vaccination coverage for pediatric vaccinations has been achieved in recent years due to the approval of the law on compulsory vaccination for kindergarten school attendance in July 2017 [13,14,15,16].

In 2020, the “Communication in Public Health” working group (WG) of the Italian Society of Hygiene, Preventive Medicine, and Public Health (SItI), created in 2018, suggested a decalogue for Italian Prevention Departments based on the WHO document on Risk Communication and Community Engagement in Response to SARS-CoV-2 [17].

The present “decalogue” of strategic actions to counteract vaccine hesitancy comes from the cognitive and formative path put in place by the “Communication in Public Health” WG, which was supported in the drafting by the “Vaccinations” and “Leadership” WGs [18].

All the participants of a roundtable, organized in September 2021 by the cited SItI working groups, had been asked to define a document that would collect the general principles in response to Italian vaccine hesitation, not only in relation to COVID-19 vaccines but to all vaccinations in the National Immunization Plan (NIP).

The initial proposal was subject to a review process by the operators of the restricted boards of the SItI working groups involved with the aim of having a corporate document that could outline a “roadmap” and define the commitment of our scientific society in combating the worrying phenomenon of vaccine hesitancy in Italy.

## 2. The Ten Actions Suggested by the Italian Society of Hygiene Preventive Medicine and Public Health

### 2.1. Establishment of a National Multidisciplinary Working Group on Vaccine Hesitancy

Just as the WHO decided to set up a subgroup of SAGE to deal with vaccine hesitancy, it was considered equally appropriate, given the immediate negative consequences of vaccine hesitancy associated with the COVID-19 vaccination campaign and all other vaccines in the National Immunization Plan (NIP), to create a group of experts in public health, sociology, and communication that could make recommendations and indications to facilitate and promote actions at the national and local levels in order to challenge vaccine hesitancy [19]. Moreover, general practitioners and pediatricians should also be involved for their key role in access to primary care and vaccination for the general population.

### 2.2. Activation of a National Monitoring/Surveillance System on Vaccine Hesitancy

In the current landscape of lifestyle monitoring tools in Italy, some surveillance systems investigate the acceptance or implementation of specific vaccinations, but no national system systematically addresses the phenomenon of vaccination hesitancy [20,21].

It is important to define a set of specific questions that would be able to detect temporal variations in the hesitancy phenomenon, including geographical differences and any other element that could help detect criticalities, and to implement direct action on the basis of the relevance and weight of the hesitancy determinants, which can change over time, space, and from vaccine to vaccine. Otherwise, the use of an innovative platform based on the filter of social media, in order to obtain the overall vaccine stance of the population in real time, should be considered [22]. The monitoring of vaccine hesitancy is also fundamental to the identification of “good practices”, which can be implemented at any useful level, including at the operational level, by vaccination services.

### 2.3. Identification and Dissemination of Tools and Methods to Measure and Understand Vaccine Hesitancy

Vaccine hesitancy is, by definition, a complex phenomenon, characterized by multiple determinants that need to be studied in a systematic way, and which require different methods of analysis.

The monitoring of knowledge, attitudes, and behavior, for example, can only cover some determinants (mainly individual determinants), but this will not be useful for understanding the relevance of some contextual determinants, which will require different and more complex methods of investigation.

As a valuable example, the SAGE working group on vaccine hesitancy developed a survey tool to assess the nature and extent of vaccine hesitancy problems with the aim of characterizing the nature and extent of vaccine hesitancy problems and of better informing the development of appropriate strategies and policies to address the concerns expressed and to sustain confidence in vaccination.

Vaccine hesitancy questions were piloted in the annual WHO–UNICEF joint report form, completed by national immunization managers globally.

Addressing vaccine hesitancy in all its complexity is a challenge and necessity that requires directed and coordinated effort [23].

### 2.4. Identification, Testing, and Dissemination of Local and National Good Practices to Counteract Vaccine Hesitancy

There is a wealth of evidence in the scientific literature on strategies and actions to combat vaccine hesitancy [24,25].

As with any public health intervention applied not only to individual patients, but to entire populations, activities aimed at combating vaccine hesitancy are not guaranteed to be effective and applicable to any given population or population subgroup.

It is extremely important to establish what has worked, is working, and can work in Italy, in the various Italian regions, and in relation to the different determinants of vaccine hesitancy across the different regions. For this reason, it is necessary to promote the transfer of good practices in every useful context to encourage testing and the evaluation of interventions in population samples or in specific contexts in order to establish the effectiveness of interventions that may be transferable to the general population.

As an example, the Council of Canadian Academies Expert Panel on Health Product Risk Communication Evaluation (2015) developed an inventory of vaccination communication toolkits in Canada. Five best practices were identified: (1) identify target audience and establish trust; (2) provide both the risks and benefits of vaccination, as most people are looking for balanced information; (3) apply only evidence-based risks; (4) improve data visualization; and (5) practice the communication toolkit before using it [26].

### 2.5. Widespread and Interdisciplinary Training on Vaccine Hesitancy

Multidisciplinary, interdisciplinary, and intersectoral training can facilitate the growth of a widespread culture and a better understanding of complex phenomena. The topic of “vaccine hesitancy” requires specific training efforts aimed at disseminating knowledge on measurement tools and effective actions to address it. It is important to educate the medical profession to interface with the phenomenon of vaccine hesitancy.

A good strategy would be to dedicate a specific course within the university curriculum of health professionals. The Oakland University William Beaumont School of Medicine organized a multifaceted interactive session with the aim of providing preclinical students with knowledge and skills to improve communication with VH patients and parents and the need for ongoing practice with respect to these VH counseling skills [27].

### 2.6. Training on Risk Communication, Community Engagement and Infodemiology

It is necessary to give more space to “infodemia”, “risk communication” and “community engagement” within the national panorama of training activities, consistent with the relevance of the phenomena (infodemia) and tools (risk communication and community engagement).

Information, misinformation, and public health are intertwined, and the WHO has dealt with issues around this intersection since its founding [28].

In response to the pressing demand for information about COVID-19, in fact, the WHO established the Information Network for Epidemics (EPI-WIN), which disseminates and amplifies evidence-based information about COVID-19, and tracks and responds to misinformation, myths, and rumors. [29].

Misinformation is increasingly sophisticated, hard to track, and emotive, and it can encourage behaviors that harm health (rejecting health interventions such as vaccines, disregarding health guidance, trying out unproven and dangerous therapies) [30].

The WHO has evolved its risk communication and community engagement approach during every major global outbreak, from smallpox to HIV/AIDS to A/H1N1pdm09 to Ebola to Zika and to COVID-19 [31].

It is also relevant to highlight that the WHO itself has made efforts to train healthcare professionals in the field of “infodemiology” and to facilitate the national dissemination of “infodemiological” knowledge.

### 2.7. Inclusion of Effective Interventions to Counteract Vaccine Hesitancy within the National Immunization Plan (NIP)

Europe is one of the regions in the world with the highest level of vaccine hesitancy, particularly with respect to vaccine safety concerns [32].

European vaccine hesitancy can also be attributed to false perceptions among the general population that vaccines do not work; a distrust of information; perceived low risks from vaccine-preventable diseases; and a lack of trust in HCWs, authorities, and pharmaceutical companies [33].

Given the complexity of vaccine hesitancy and the limited evidence available on how it can be addressed, the strategies identified should be carefully tailored to the target population, its reasons for hesitancy, and the specific context [34].

In order to ensure a uniform application of interventions to contrast vaccine hesitancy, throughout Italy in particular, it is necessary to propose a change in the National Immunization Plan (NIP) that provide for this activity. In addition, it is necessary to propose, in parallel, methods and indicators to evaluate the effectiveness of the implementation of such interventions in different population targets, including healthcare professionals [35,36].

### 2.8. Promoting the Establishment and Growth of a Community of Practice and Research in the Field of Vaccine Hesitancy

The Strategic Advisory Group of Experts on Immunization (SAGE) advises the WHO on overall global policies and strategies [37].

In particular, the SAGE working group on vaccine hesitancy (WG) proposed a definition of hesitancy and a model to classify the factors influencing the behavioral decision to accept a vaccine [38].

Many research groups in Italy are working on vaccine hesitancy, but there is a lack of a real community of practice, which, on the contrary, should be made up of all the research groups interested in vaccine hesitancy to facilitate coordination, collaboration, and synergies.

### 2.9. Promoting Collaborations between Scientific Societies

Italian scientific societies, guided by the SItI together with other scientific representations of the mathematical sciences, digital health, social and behavioral sciences, communication and journalism, and marketing should be able to collaborate through memoranda of understanding, consensus documents, and shared guidelines in order to measure and contrast vaccine hesitancy more effectively with respect to the complexity of the problem and the need for an interdisciplinary approach.

In order to tackle vaccine hesitancy, the multidisciplinary team should act by recognizing and implementing the “5C” (confidence, complacency, convenience, communications, and context) model of factors influencing vaccine hesitancy [39].

### 2.10. Promoting Knowledge from the Behavioral Sciences

Knowledge from the behavioral sciences is useful in identifying good practices to promote drivers of vaccine acceptance, in particular: a favorable context (quality of experience, time spent, costs, personal benefits); social influences (vaccines as a social norm; training of health personnel; presumptive communication; opinion leaders and influencers); and motivation (social benefits of vaccination, economic benefits, family benefits, etc.).

It is therefore necessary to disseminate knowledge on the use of the different behavioral models in different population settings [40].

## 3. Conclusions

Vaccine hesitancy could compromise the success of vaccination programs. Given its complexity, it is therefore necessary to address the problem of vaccine hesitancy in a broader and crosscutting way through the implementation of evidence-based strategies at various levels, the promotion of the transfer of good practices to every useful context, and the testing of interventions on population samples or specific settings in order to establish the effectiveness of actions that may be transferable to the general population.

In this regard, scientific research in communication has improved a lot in the last few years with respect to investigating new values to find the tools that best meet the final user’s needs [41]. As communication in an emergency will obviously need to be timely and to adapt continuously to quick changes, communication tools should be developed on a fast track based on existing practices that need to be finetuned to the specific heath emergency.

Traditional study designs should remain central to the development of communication strategies and should be coupled with small experiments to be conducted in a short timeframe in different settings. As suggested for the vaccine communication strategy, the development process should be articulated in the following steps: listening to public concerns on vaccines; informing the content of the messages; cyclically monitoring the effectiveness of the campaigns and refining the communication strategies based on feedback and performance; and then measuring and evaluating the outcomes [42].

The present document suggests ten undeferrable strategies that could be implemented at the national and local levels in Italy, and that could be borrowed by other European countries in order to counteract vaccines hesitancy with a systematic and organic approach.

## Data Availability

The study did not report any data.

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
