# Peer review of "Ten Actions to Counteract Vaccine Hesitancy Suggested by the Italian Society of Hygiene, Preventive Medicine, and Public Health"

_vaccines, 2022, doi:10.3390/vaccines10071030_

Round 1

Reviewer 1 Report

I was invited to revise the paper entitled "The Ten Actions to counteract vaccine hesitancy of the Italian Society of Hygiene, Preventive Medicine and Public Health". It was a document presented by a section of the Italian Public Health Society aiming to address the most important issues to face off vaccine hesitancy. 

I want to congratulate with Authors for the excellent work and the well written paper.

The paper is well written and deeply describes the topic of VH, citing important references from ECDC and WHO. 

It is well known that VH phenomenon spreads across all over Europe and Italy was one of the country most involved in this issue. 

About the 10 suggested actions, I have some minor observations:

- Point 1: about the multidisciplinary group of expert, in my opinion also general practitioner and paediatrician should be involved. Parents have frequently access to primary care and these experts could help in addressing  this point;

- Point 7: in my opinion also intervention focused on VH among healthcare workers should be included. VH among HCW is a public health issue, particularly in Italy, as reported in some previous research papers: (10.3390/v13030371; 10.3390/vaccines8020248; 10.15167/2421-4248/jpmh2019.60.1.1097).

Author Response

Reviewer #1

I was invited to revise the paper entitled "The Ten Actions to counteract vaccine hesitancy of the Italian Society of Hygiene, Preventive Medicine and Public Health". It was a document presented by a section of the Italian Public Health Society aiming to address the most important issues to face off vaccine hesitancy. 

I want to congratulate with Authors for the excellent work and the well written paper.

The paper is well written and deeply describes the topic of VH, citing important references from ECDC and WHO. 

It is well known that VH phenomenon spreads across all over Europe and Italy was one of the country most involved in this issue. 

Dear reviewer,

Firstly, thank you for appreciating our manuscript. We hope that this revised version could further  improve the article in accordance with your useful suggestion.

Your comments were considered with attention and a point by point answer to your remarks and questions was reported below.

Comment: About the 10 suggested actions, I have some minor observations:

- Point 1: about the multidisciplinary group of expert, in my opinion also general practitioner and paediatrician should be involved. Parents have frequently access to primary care and these experts could help in addressing  this point.

Answer: Your suggestion is absolutely correct and the two figures were added in the manuscript

C: In my opinion also intervention focused on VH among healthcare workers should be included. VH among HCW is a public health issue, particularly in Italy, as reported in some previous research papers: (10.3390/v13030371; 10.3390/vaccines8020248; 10.15167/2421-4248/jpmh2019.60.1.1097).

Answer: The suggestion is absolutely correct. The consideration and the references in accordance were added in the text.

Reviewer 2 Report

Public health agencies, and the authors here representing that perspective, have a job to do. They need to promote health in the population but sometimes encounter pockets of resistance or other barriers to implementing best practices. The article calls for action. The particular actions seem fine and well-intended. As for as that goes, the authors have done well in provided a set of related activities. Most of the remainder of my comments, however, ask for some attention to a broader perspective that respects (more) the nature of hesitancy, its appropriateness in some contexts, and the boundaries or limits to full compulsion or extraordinary pressure on sovereign citizens. 

The article directs itself to Italy and other European countries. The situation of Democratic regimes is sometimes harder than Totalitarian regimes because they tend to respect more freedom of thought and action among citizens to make decisions that are right for them. A problem with infectious disease is that such decisions are subject to externalities or spillover effects whereby an infected and ill person can pass along the illness and make life worse for others. Public health agencies probably often represent the perspective of an average citizen, whereby it is advantageous from a risk-benefit perspective for all “others” to be vaccinated. “They” bear the risks of side effects, etc. but convey the community benefit of lower externalities to others. Still, there are side effects to vaccines whether they be variable or for a time unknown.

Using the term “hesitancy” suggests a one-sided view of the educational campaigns and other means of allowing individuals to make informed decisions for themselves but also considering the implications for others. Given such high vaccination rates in Italy for Covid, it would seem that is not particularly a crisis. Therefore, this article might articulate a broader framework that homes in on situations in which “hesitancy” is knowingly unwarranted but is sensitive to situations in which hesitancy is understandably acceptable or even appropriate. Are the authors espousing a propaganda agenda, or an educational public service? Do the authors wish to take a stance regarding respect for the role of “personal, cultural, or religious” reasons that could motivate someone to decline a vaccine? In other words, how do those factors enter the agenda alongside scientific uncertainty or misunderstanding? The article says: “Satisfactory vaccination coverage for paediatric vaccinations has been achieved in recent years thanks to the approval of the law on compulsory vaccination in July 2017 [13-16].” Does this suggest that the authors universally favor (“thank”) legal force and other compulsions to overcome “hesitancy?” This question is particularly important given the article says: “…were asked to define a document that would collect the general principles in response to Italian vaccine hesitation, not only in relation to covid-19 vaccines but to all vaccinations in the National Immunization Plan (NIP).”

To what extent do the authors wish to promulgate guidelines for determining when hesitancy is the right decision, affirming that, and distinguishing that from a goal of universal vaccination?

Distinguishing those alternate scenarios can be difficult for many parties, including individuals and families, especially when the “appropriate message” is provided along with strong financial or political motivations among the advocates, including pharmaceutical companies, organized medicine, or political ideologues. The article states: “Information, misinformation, and public health are intertwined, and WHO has dealt with issues around this intersection since its founding [26].”

The article states: “There is a wealth of evidence in the scientific literature on strategies and actions to combat vaccine hesitancy.” but strangely provides no citations at that point. This seems like a major gap because the article is a call for action that seemingly rests on grounded science.  What is new here, and what deviates from the “wealth of evidence?”

The article states: “European vaccine hesitancy can also be attributed to a perception that vaccines do not work, distrust of information, perceived low risks of vaccine-preventable diseases, and to a lack of trust in HCWs, authorities, and pharmaceutical companies [31].”  This statement seems to be a reversal of cause and effect.  Its structure implies that hesitancy causes the reasons for hesitancy rather than the other way around.

Author Response

Public health agencies, and the authors here representing that perspective, have a job to do. They need to promote health in the population but sometimes encounter pockets of resistance or other barriers to implementing best practices. The article calls for action. The particular actions seem fine and well-intended. As for as that goes, the authors have done well in provided a set of related activities. Most of the remainder of my comments, however, ask for some attention to a broader perspective that respects (more) the nature of hesitancy, its appropriateness in some contexts, and the boundaries or limits to full compulsion or extraordinary pressure on sovereign citizens. 

The article directs itself to Italy and other European countries. The situation of Democratic regimes is sometimes harder than Totalitarian regimes because they tend to respect more freedom of thought and action among citizens to make decisions that are right for them. A problem with infectious disease is that such decisions are subject to externalities or spillover effects whereby an infected and ill person can pass along the illness and make life worse for others. Public health agencies probably often represent the perspective of an average citizen, whereby it is advantageous from a risk-benefit perspective for all “others” to be vaccinated. “They” bear the risks of side effects, etc. but convey the community benefit of lower externalities to others. Still, there are side effects to vaccines whether they be variable or for a time unknown.

Dear reviewer,

Thank you for the opportunity to revise our manuscript. Your consideration are quite shareable, the topic is an absolute priority for Public Health Authorities and the attention that you required in you comments and suggestions are useful for our perspective.

We also would thank you for appreciating the contents and we hope that this revised version could improve the article and may have solved the major problems raised by your revision.

Your useful comments were considered with attention and a point by point answer to your remarks and questions was reported below.

Comment: The article says: “Satisfactory vaccination coverage for paediatric vaccinations has been achieved in recent years thanks to the approval of the law on compulsory vaccination in July 2017 [13-16].” Does this suggest that the authors universally favor (“thank”) legal force and other compulsions to overcome “hesitancy?”

A: Thank you for your observation. The authors by this statement in no way intend to express satisfaction with the achievement of desirable vaccination coverage through legal obligations. Instead, the aim of this manuscript is to construct actions that, while respecting the personal, cultural or religious factors that influence vaccination hesitation, can convince patients to make an informed and correct decision for their health. We modified the sentence in accordance.

Comment: To what extent do the authors wish to promulgate guidelines for determining when hesitancy is the right decision, affirming that, and distinguishing that from a goal of universal vaccination?

Distinguishing those alternate scenarios can be difficult for many parties, including individuals and families, especially when the “appropriate message” is provided along with strong financial or political motivations among the advocates, including pharmaceutical companies, organized medicine, or political ideologues. The article states: “Information, misinformation, and public health are intertwined, and WHO has dealt with issues around this intersection since its founding [26].”

A: Thank you for your suggestion. We will try to convey a broader view of the problem. Surely what should guide one to vaccination should be the evidence of scientific data. Regardless of which source delivers the 'appropriate message'.

Comment: The article states: “There is a wealth of evidence in the scientific literature on strategies and actions to combat vaccine hesitancy.” but strangely provides no citations at that point. This seems like a major gap because the article is a call for action that seemingly rests on grounded science.  What is new here, and what deviates from the “wealth of evidence?”

A: In accordance with your useful suggestions, we will provide for sure some citation at that point. Anakpo G, Mishi S. Hesitancy of COVID-19 vaccines: Rapid systematic review of the measurement, predictors, and preventive strategies. Hum Vaccin Immunother. 2022 Jun 17:2074716.

Ames, Heather Mr et al. Parents' and informal caregivers' views and experiences of communication about routine childhood vaccination: a synthesis of qualitative evidence. The Cochrane database of systematic reviews vol. 2,2 CD011787. 7 Feb. 2017.

Comment: European vaccine hesitancy can also be attributed to a perception that vaccines do not work, distrust of information, perceived low risks of vaccine-preventable diseases, and to a lack of trust in HCWs, authorities, and pharmaceutical companies [31].”  This statement seems to be a reversal of cause and effect.  Its structure implies that hesitancy causes the reasons for hesitancy rather than the other way around.

A: As correctly suggested we will modify the structure of the sentence in order to make it clearer.